# COVID-19 spread, detection, and dynamics in Bogota, Colombia

Rachid Laajaj [1✉], Camilo De Los Rios[2], Ignacio Sarmiento-Barbieri [1], Danilo Aristizabal [1], Eduardo Behrentz [1], Raquel Bernal[1], Giancarlo Buitrago [3,4], Zulma Cucunubá[5,6], Fernando de la Hoz[3], Alejandro Gaviria[1], Luis Jorge Hernández[1], Leonardo León[1], Diane Moyano[7], Elkin Osorio[7], Andrea Ramírez Varela[1], Silvia Restrepo [1], Rodrigo Rodriguez[7], Norbert Schady[8], Martha Vives[1] & Duncan Webb [9]

Latin America has been severely affected by the COVID-19 pandemic but estimations of rates of infections are very limited and lack the level of detail required to guide policy decisions. We implemented a COVID-19 sentinel surveillance study with 59,770 RT-PCR tests on mostly asymptomatic individuals and combine this data with administrative records on all detected cases to capture the spread and dynamics of the COVID-19 pandemic in Bogota from June 2020 to early March 2021. We describe various features of the pandemic that appear to be specific to a middle income countries. We find that, by March 2021, slightly more than half of the population in Bogota has been infected, despite only a small fraction of this population being detected. The initial buildup of immunity contributed to the containment of the pandemic in the first and second waves. We also show that the share of the population infected by March 2021 varies widely by occupation, socio-economic stratum, and location. This, in turn, has affected the dynamics of the spread with different groups being infected in the two waves.

[1] University of Los Andes, Bogota, Colombia. [2] Inter-American Development Bank, Washington, D.C., USA. [3] Universidad Nacional de Colombia, Bogota, Colombia. [4] Hospital Universitario Nacional de Colombia, Bogota, Colombia. [5] Imperial College London, London, UK. [6] Universidad Pontificia Javeriana, Bogota, Colombia. [7] Secretaria de Salud de Bogota, Bogota, Colombia. [8] World Bank, Washington, D.C., USA. [9] Paris School of Economics, Paris, France. ✉email: r.laajaj@uniandes.edu.co

As of March 8th 2021, the World Health Organization (WHO) has declared 116 million detected cases of COVID-19 worldwide. 43% of these detected cases have occurred in lower and middle income countries[1], a group which comprises 81% of the global population. Taken at face value, these statistics would imply that COVID-19 is more prevalent in high income countries. But recent evidence points towards high levels of infection in many low and middle income countries[2,3], with some of the highest rates of infection expected to occur in Latin American countries[4]. Moreover, high rates of infection in low-income areas may not be well reflected in recorded cases; the ability to detect cases may vary dramatically, for example ranging from an estimated one in four cases in Philadelphia, USA, to one in 621 cases in Kenya[3,5–8].

Research documenting the differential impact of the COVID-19 pandemic across occupational or socioeconomic groups is largely limited to high income countries[9] with some attempts in middle income countries that are suggestive but constrained by the limited data availability[10]. However, low and middle income countries have specific features that are likely to affect the transmission patterns of the virus. For example, workers may be less able to work remotely[11,10]; rates of informal work may be higher, with less sick leave; institutions may have lower capacity to implement quarantine, isolation, testing, and tracing measures; use of public transportation may be higher; and housing conditions may be more crowded. It is therefore important to study the spread of the pandemic in such context in order to better inform local authorities' decisions for targeted interventions or restrictions.

## Results

**Data and estimation.** Our primary data comes from the CoVIDA project, a sentinel community-based surveillance initiative led by the University of Los Andes that gathered information on an average of 1552 RT-PCR tests per week in Bogota over a 9-month period (see "Methods" Section for a description of the test). We invited an adult population to be tested, most of whom were asymptomatic at the time of taking the test. We over-sampled occupations that were expected to be most exposed to the virus, such as security guards, transportation workers, and health workers. The first half of the sample was randomly drawn from lists of participants obtained through agreements with a number of local partners, mostly composed of large employers. Many of these lists constitute a large share of a given occupation, ensuring that our samples were representative within each occupation. The second half was a convenience sample based on our "public campaign": a widely disseminated public invitation for free testing. Results are qualitatively similar when restricting to the sample that is randomly drawn from lists (see Supplementary Table 1 and Supplementary Fig. 3). We also use administrative data collected by the Health Secretary of Bogota (HSB, Secretaria de Salud de Bogota in Spanish) that covers all the reported cases in Bogota from the beginning of the pandemic on January 23rd, 2020 up until February 14, 2021.

In our main estimations, we reweight observations by occupation in order to ensure that our results are representative of the whole population of Bogota. We exclude individuals with symptoms and those who are known to have been in contact with an infected person, resulting in a main sample with 42,164 observations. This conservative assumption allows us to avoid any bias resulting from these individuals (who typically have higher prevalence rates) being more likely to seek or accept testing[12]. We also estimate the daily case incidence using the positivity rate for CoVIDA tests. This calculation converts from prevalence to incidence by assuming that individuals can be tested positive for a mean period of 17 days (a figure based on the estimated

sensitivity of the RT-PCR tests used in the study[13]). For more details on the sample, calculation methods and robustness, see the "Methods" Section.

**From the tip of the iceberg to its actual size.** We use the CoVIDA database to estimate total cases in Bogota (the actual size of the iceberg) and compare it to the number of detected cases from the HSB (the tip of the iceberg). As shown in Fig. 1, our estimation of the cumulative number of cases per 100,000 inhabitants is highly consistent with the positivity rate obtained in a seroprevalence survey administered by the National Health Institute of Colombia (NHI) between October 26th and November 17th, 2020.

We draw multiple lessons from these figures. First, by March 3rd 2021, 53% [95% CI: 45–62%] of the population in Bogota had been infected. This is in line with other studies showing total incidence rates nearing 50% in Latin America[2,4,14], compared to rates of 1 to 15% in high income countries[7,15,16]. Second, only 8% of the population had been tested positive by the end of January, implying that approximately one in every 6.4 cases is detected [95% CI: 5.4–7.5]. This is a relatively high detection rate compared to other low and middle income countries[3,5,6,8]. However, detection rates vary significantly even within Bogota, ranging from one in 10.1 for the lowest socioeconomic stratum to one in 5.9 for the highest strata (see Supplementary Table 3).

Third, other studies have found that cases typically start to decrease between 1 and 4 weeks after the beginning of generalized quarantine in European countries, US, China, Iran and Turkey[17,18]. By contrast, while the quarantine that started on March 24th 2020 in Bogota helped to "flatten the curve" and avoid overcrowding of hospitals, the number of daily new cases continued to increase. We estimate that the reproduction number during the early phases of quarantine, when the fraction of the population that is susceptible (S) is assumed to be ~1, is $R_q = 1.22$ [95% CI: 1.17–1.27]. It appears to be stable during the first months of the quarantine (see Supplementary Note 2 or our companion paper[19]). Since the effective reproduction number $R_e$ can be found by multiplying $R_q$ by the proportion of susceptible individuals in the population S, a simple model in which $R_q$ remains constant would require S to be below 0.82 for $R_e$ to fall below one. This is highly consistent with a downward trend of the first peak that started by the end of July, when the share of infected reached almost 20% of the population. In short, the quarantine alone was insufficient to curb the increase in daily new cases until a significant fraction of the population was immune.

Condensing the story of the pandemic in Bogota into a simple framework, we attribute most variation in transmission to two dominant mechanisms: the number of contacts and susceptibility. These are well reflected in the calculation of the effective reproduction number $R_e = S \times n \times SAR$, where $n$ is the average number of contacts for an infected individual during their infectious period, $SAR$ is the secondary attack rate (the probability of infection conditional on contact) and S is the share of the population that is susceptible to the disease (never infected, if infection is associated with immunity). Mobility levels in Bogota went down considerably at the beginning of the quarantine, for example through less time commuting or at work, likely leading to large reductions in $n$ (see Supplementary Fig. 6). However, this mobility increased again over time as a result of the growing need for people to resume economic and social activities, and a progressive loosening of the quarantine. The number of contacts, and with it, $R_e$, thus increased over time. On the other hand, as a consequence of the growing share of the population having been infected shown in Fig. 1, the share of susceptible individuals went down over time, slowing down the expansion of

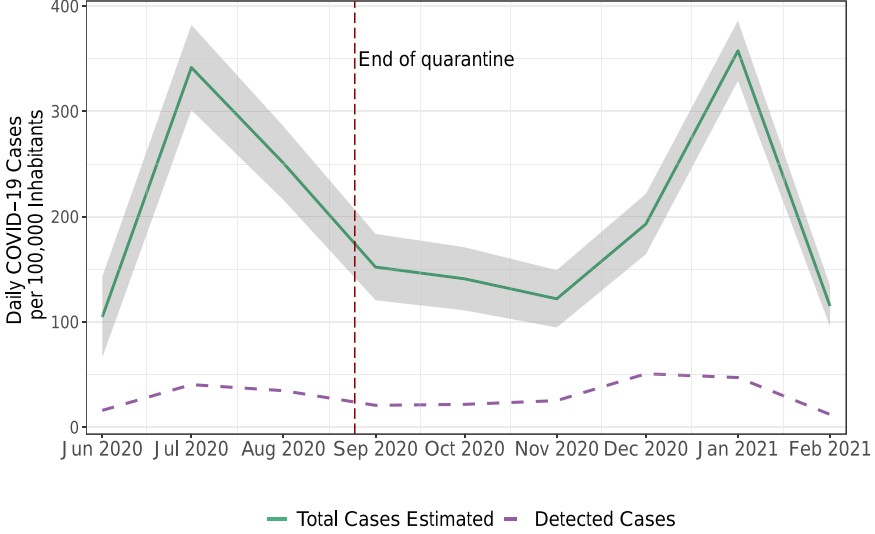

(a) Daily Cases per 100,000 Inhabitants

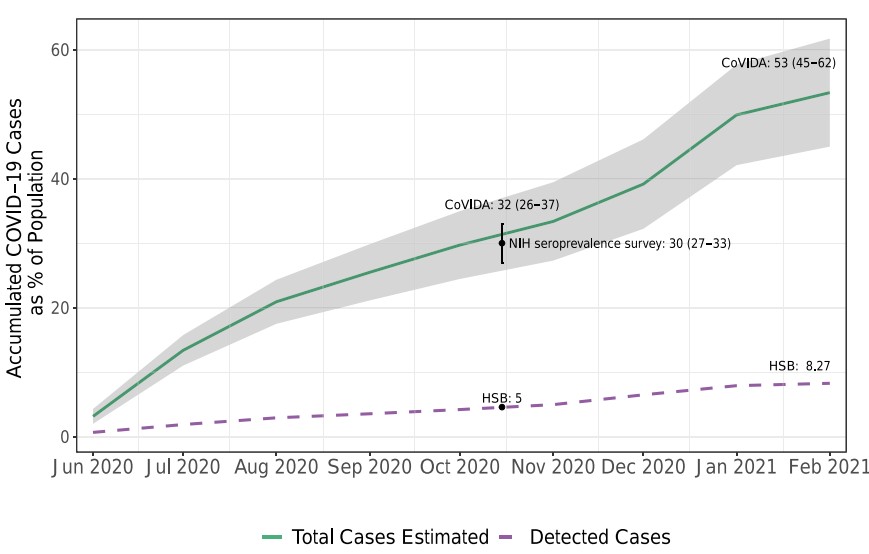

(b) Accumulated Cases as % of Population

**Fig. 1 Daily new cases of COVID-19 per 100,000 inhabitant and accumulated cases as % of population. a** Total daily COVID-19 cases per 100,000 inhabitants based on CoVIDA data and detected cases based on data from the Health Secretary of Bogota (HSB). The vertical dashed line marks the end of quarantine on August 25, 2020. **b** Cumulative cases as % of Bogota's population. In black the point estimate and 95% confidence interval of the test positivity rate from a seroprevalence survey run by the National Health Institute of Colombia (NHI)[30]. Green solid line in (**a**) represents the weighted estimates of the number of daily cases per 100,000 inhabitants for each month ($n = 42,164$). The purple dashed line in (**a**) represents the number of cases per 100,000 inhabitants reported by the HSB. Green solid line in (**b**) represents the weighted accumulated estimated daily number of cases in (**a**) % of population. Weights are based on workers' occupation to be representative of Bogota's population. Shaded area denotes 95% confidence intervals. Dashed purple line represents the accumulated number of cases as % of population reported by HSB.

the virus. After the end of the full quarantine, the two forces seemed to be balanced in the period from September to November. Subsequently, the second wave may have been driven by an increase in the number of contacts during festivities.

**An uneven distribution of infections**. In Fig. 2, we analyze heterogeneity in infections by occupation, socioeconomic strata, and locality. As expected, occupations that are most likely to be able to work remotely have some of the lowest incidence rates, while occupations that require physical interactions tend to be at the top of the distribution[20]. Accordingly, security guards, construction workers, shopkeepers, taxi drivers, public transportation workers, military, and police were all defined as priority populations for testing by the CoVIDA project because of their exposure to multiple contacts. Less anticipated were the high levels of infection among homemakers and the unemployed,

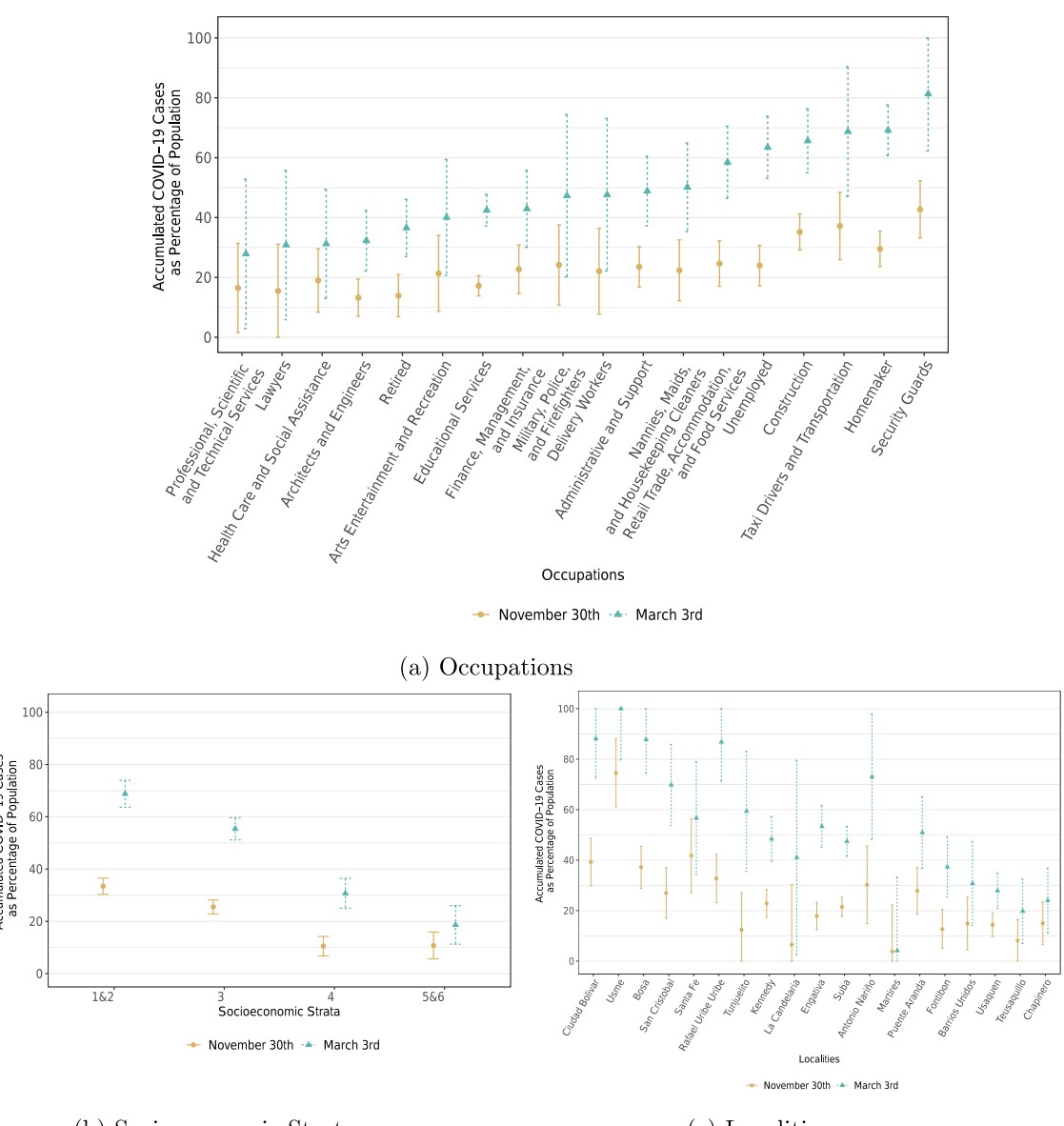

(a) Occupations

(b) Socio-economic Strata                                          (c) Localities

**Fig. 2 Estimated accumulated COVID-19 Cases as a percentage of population by occupation, socioeconomic stratum, and district: june 1st—November 30th and June 1st—March 3rd. a** By occupations. **b** By socioeconomic strata (**c**) by Bogota's districts, sorted by district's mean stratum. Dots represent weighted point estimates of accumulated cases between June 1st–November 30th (brown dots) and and June 1st–March 3rd (teal triangles) ($n = 42,164$). Error bars denote 95% confidence intervals. Observations are weighted by occupation to be representative of the Bogota population. Supplementary Tables 4, 6, 7 show the sample decomposition by occupations, Bogota's districts, and socioeconomic stratum respectively.

which may be explained by the low socioeconomic status of these groups. The results also suggest quite different dynamics among the various groups. For example, security guards, transportation workers and tellers had already reached high levels of infection by the end of November. These occupations were considered "essential workers", exempt from the full quarantine. By contrast, most of the infections for shopkeepers, construction workers, babysitters and house workers occurred between November and February, which is consistent with the fact that they had stronger restrictions to commute and work on site early on in the pandemic (more details on quarantine restrictions by occupation in Supplementary Table 4).

Figure 2b displays a similar heterogeneity in infections by socioeconomic stratum, a classification used in Colombia as a proxy for the household's socioeconomic conditions. Neighborhoods are categorized into one of 6 levels, where 1 is the poorest

and 6 is the wealthiest. To gain power, we pool together strata 1 and 2 and strata 5 and 6. The relationship between socioeconomic condition and infection is monotonic; the poorest strata were four times more likely to have been infected than the wealthiest ones. Lower strata were hit particularly hard during the first period, whereas middle and higher strata had a large share of cases occurring during the second period. This is consistent with the observation that higher strata were more likely to be subject to the full quarantine in the earlier period, meaning that they developed less immunity and started the second period with a high share of susceptible individuals. Figure 2c shows an even greater level of heterogeneity between districts (Bogota includes 20 districts, which are the largest geographical division). The virus spread more widely and more quickly in districts that are poorer (see Supplementary Figs. 9, 10 for maps of the positivity rate, infections and socioeconomic status by districts).

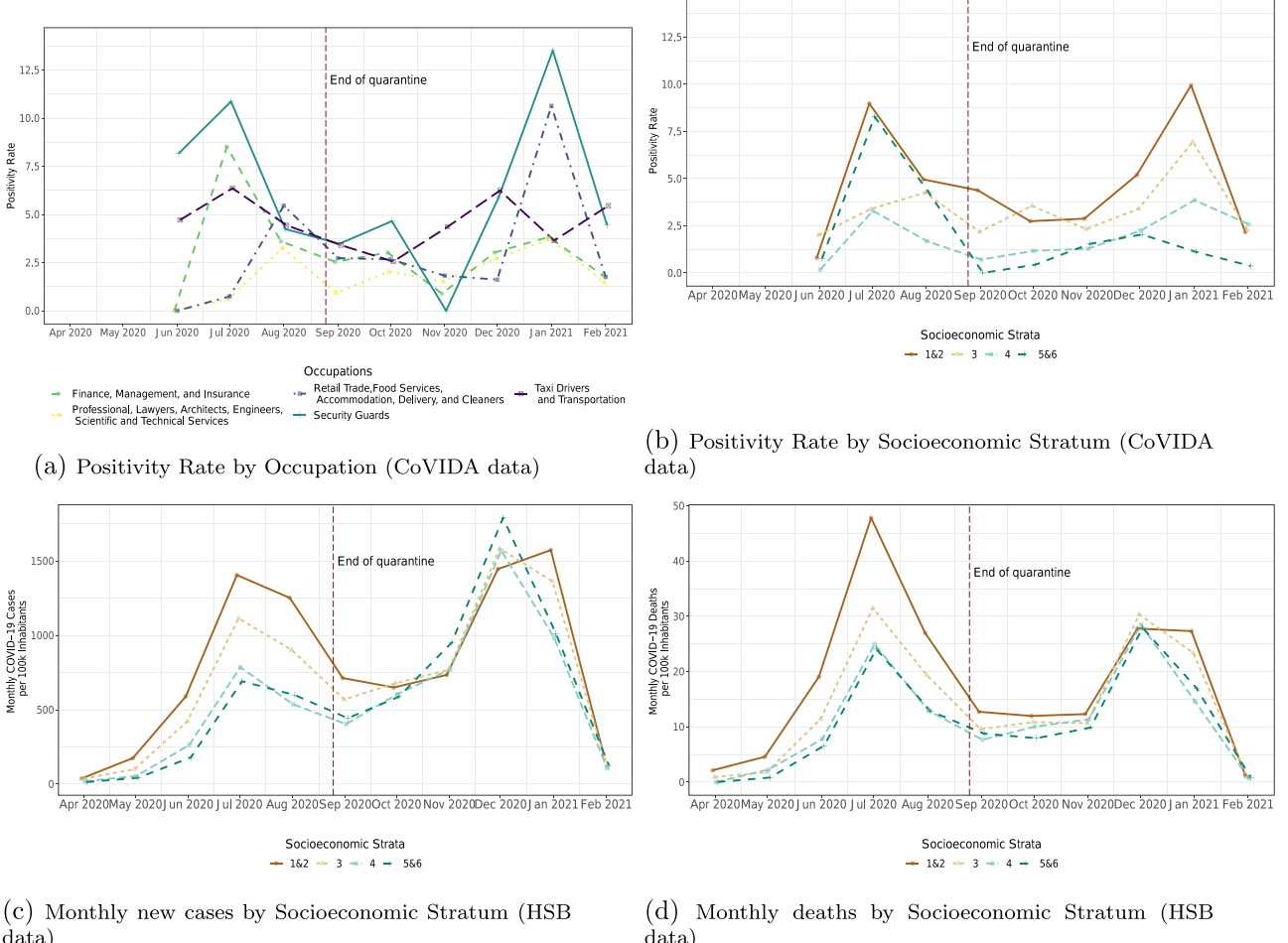

(a) Positivity Rate by Occupation (CoVIDA data)

(b) Positivity Rate by Socioeconomic Stratum (CoVIDA data)

(c) Monthly new cases by Socioeconomic Stratum (HSB data)

(d) Monthly deaths by Socioeconomic Stratum (HSB data)

**Fig. 3 Monthly dynamics.** The figure shows weighted monthly COVID-19 positivity rates (**a**, **b**) from CoVIDA data and monthly COVID-19 cases per 100,000 inhabitants (**c**, **d**) from the Health Secretary of Bogota (HSB). The vertical dashed line marks the end of quarantine on August 25, 2020. Weights are based on workers' occupation to be representative of Bogota's population. To maintain comparability, (**d**) uses the date of the positive test of deceased individuals on the horizontal axis. See Supplementary Fig. 7 for 95% confidence intervals.

**Different dynamics**. After finding that COVID-19 exposure varies by group and over time, we analyze how this translates into different dynamics of infections across the various groups. Figure 3a illustrates how much the dynamics can differ by occupational group. For example, the group of workers most able to work remotely (e.g., lawyers, engineers, and scientists) was able to maintain low infection rates throughout the study period. By contrast, tellers contributed mostly to the first peak, shopkeepers to the second peak, and security guards to both. (The dynamics of all groups appear in Supplementary Fig. 8.)

We capture variations by stratum using a number of different data sources. Figure 3b uses CoVIDA data, which is closer to a random sample of the population, but has a smaller sample size, reducing statistical power at higher levels of disaggregation by group and time. Figure 3c uses reported cases from the HSB data, with a large sample size but possible bias due to differential detection rates across strata and time. We also report monthly COVID-19 deaths (Fig. 3d), which is likely to be subject to fewer selection and sample size issues.

The data from the HSB show a very clear ordering during the first wave, with a strong decrease in infection rates as income increases (Fig. 3c). Interestingly, this relationship is not observed at all in the second wave. Instead, all strata exhibit similar rates of daily new cases from COVID-19, and if anything, strata 5 and 6 have the highest rates of daily new cases reported at the beginning

of the second wave. However, this is not reflected in the CoVIDA data nor in deaths associated to COVID-19 in official data (Fig. 3b, d). This puzzle seems to be driven by the fact that, between July 2020 and January 2021, the share of cases that were detected grew from 5% to 52% in strata 5 and 6, while it remained relatively stable in the poorest strata (Supplementary Table 3). The differences in detection dynamics across strata may be a consequence of the government program PRASS ("Programa de Pruebas, Rastreo y Aislamiento Selectivo Sostenible"), the Colombian adaptation of the Test, Trace and Isolate strategy, which began in 08/12/2020[21]. The program decentralized the testing process so that health care providers were responsible for detecting the cases of their affiliated members. Although in Bogota 97% of individuals are affiliated to a health care provider, the quality of service and appointment wait times vary greatly and is highly correlated with income levels. In keeping with this, the delegation of testing to providers appears to have dramatically increased the detection rate among the rich. This finding highlights a potential trade-off between a more centralized system that was on average less efficient but more equitable, and a decentralized system that led to improvements but only in the wealthiest strata.

By relating the recorded COVID-19 fatalities to the estimated cases from the CoVIDA data, we find an infection fatality ratio of 0.34%, a value that is plausible in a relatively young population[4].

Interestingly, recorded new daily cases are greater in the first wave than in the second one, but this is not the case if we use the CoVIDA data or registered deaths. Hence, part of the increase in detected cases is driven by an increase in detection rate over time (Supplementary Table 3).

## Discussion

A rich combination of primary and administrative data allows us to analyze the spread of COVID-19 in Bogota. Caution should be used when interpreting the results, since we are unable to use a perfectly representative sample. However, the congruence between our results and the independent serology survey implemented in the middle of our study period lends credibility to our analysis. This is one of the most extensive studies of COVID-19 on asymptomatic participants, filling an important gap in our current knowledge. Moreover, the rich primary data provides a depth that is unique in such a context, in particular allowing us to identify potential biases in official data, and to analyze which occupational and socioeconomic groups are most affected by virus spread and how this varies over time.

Compared to Bogota, the infection rates seen in the rest of Colombia have been even higher (estimated by the serology tests of the National Health Institute), with lower rate of detection (see Supplementary Fig. 11). In general, other areas of Colombia are less socioeconomically developed and have poorer institutional and laboratory capacity. This finding is thus in line with our result within Bogota that shows that lower economic strata have higher rates of infection and lower detection rates. A large part of Latin America is thus likely to have infection rates that are much higher than the level that would be suggested by their small number of detected cases.

We estimate that, as of March 3rd 2021, about 53% [95% CI: 45–62%] of the population has been infected. The implications of this result depend on the duration of the immunity that is gained from a prior infection, a question that is still the subject of some debate. Recent evidence indicates that durable immunity against secondary COVID-19 disease from the same variant is likely for most individuals[22–24]. However, the first P.1 (Brazilian) variant of SARS-CoV-2 in Bogota was detected on March 13th, and new evidence shows that previous non-P.1 infection provides less protection against infection with a P.1 lineage[25]. Results from Manaus, Brazil in early 2021 indicate that recent waves of the virus have been driven by new variants[26]. Analogously, new variants be a key explanatory factor for the magnitude of the third wave in Bogota. A better understanding of the spread and immunity response of the new variants is a growing priority in such contexts.

Given such complexities, a 53% level of infection should not be assumed to imply a 53% immunity in Bogota. Immunity has contributed to reduced transmission among some populations, but this effect is weakened by the partial loss of immunity when facing new variants. This leaves a major role for vaccines: the speed of program roll-out and vaccine effectiveness against new variants will both be key determinants of the country's recovery.

Our work shows that the high level of economic inequalities in a megacity in Latin America translates into inequalities in infection among different groups. (In a companion paper[19], we attempt to understand the key channels that drive this inequality in infections between different socioeconomic groups). We go beyond showing static differences between groups and describe how they vary over time. Understanding these dynamics is key for appropriate targeting of interventions. For example, in an active surveillance testing initiative like CoVIDA, targeting the groups with the highest infection rates will require targeting groups that are poor and have high occupational exposure during the first wave. But in the second wave, identifying populations with greater levels of exposure becomes more difficult, because the greater levels of exposure of some groups tends to be compensated by more immunity. Populations that have recently resumed their economic activity, such as teachers in the coming months, may combine low immunity and high exposure, and thus be at the greatest risk.

## Methods

**The RT-PCR test used and its reliability.** The tests to be used in this study are based on the reverse transcriptase technique—PCR or one-step RT-PCR (the U-TOP™ COVID-19 detection kit). This kit was validated in the facilities of the Public Health Laboratory of the Secretaria Distrital de Salud, finding that the kit allows detection of the COVID-19 virus, with results comparable to the Charité technique, Berlin (Rev, January 13, 2020).

**Data description.** The CoVIDA project was designed to help contain the spread of SARS-CoV-2 through active surveillance among mostly asymptotic individuals and to provide a range of information that differs from the self-selected symptomatic individuals tested in health facilities. The sample includes 59,770 RT-PCR tests of SARS-CoV-2 on 55,078 different individuals in Bogota from the beginning of June 2020 to March 3rd, 2021. At the time of registration, individuals were surveyed to capture various characteristics, including occupation, socioeconomic strata, and address.

Two main strategies were employed to recruit participants, and approximately one half of the total sample comes from each strategy. First, through 74 agreements with institutions and companies, we obtained long lists that we used to contact and invite participants. Most lists were specific to a given occupation, based on the employees of a large company or on a list of individuals who were signed up to a specific mobile app. We also used some lists of residents based on beneficiaries of social programs. We randomly selected participants from the lists and contacted them to invite them to be tested for free. The total population of all lists covers 20% of the population in Bogota. This means that it is relatively close to population-based sampling, but with an overrepresentation of some occupations that were prioritized in the CoVIDA project, in particular because they were expected to be more exposed (which is why we reweight by occupation to maintain representativity of actual occupations in Bogota, as explained below).

The second source of participants' identification comes from public announcements made by the CoVIDA team through various communication channels to invite people to be tested, stating explicitly that the invitation is open to those that are asymptomatic. The share of participants per month from each source is presented in Supplementary Table 2 and raw descriptive statistics by subgroups are provided in Supplementary Tables 4–9.

Ethics approval was obtained from the ethics committee of Universidad de los Andes (Act number 1278 of 2020). The ethics committee approved that the participants would receive the information via telephone and give their verbal consent, in order to comply with physical distancing and limit the restriction for a study is part of a public health surveillance strategy implemented jointly with the Health Secretary of Bogota.

Our second database comes from administrative records, collected by the Health Secretary of Bogota (HSB, in Spanish the Secretaria de Salud de Bogota), that cover the universe of cases of Bogota residents that have been tested positive to SARS-CoV-2 by any laboratory using an RT-PCR test, starting from the beginning of the pandemic (January 23rd, 2020) until February 14th, 2021. All laboratories in Bogota must report any positive test to the HSB, which in turn reports it to the National Health Institute that provides national statistics used by the WHO. This administrative data also comes with basic socioeconomic characteristics from a form that is a mandatory part of the institution's report when recording a positive case to the HSB.

**Estimation of positivity rates.** To obtain unbiased estimates of the true positivity rate, we need that, within each occupation, the likelihood of being tested is not systematically correlated with the likelihood of being positive within each occupation. This requires that people in the invited occupation lists are not significantly different from other non-invited lists. For example, when we have an agreement with one of the main taxi companies in Bogota, who shared a list of all its affiliated workers, we require that those workers are not systematically different from other taxi drivers. This is plausible because inclusion on the list is not an individual decision of the person invited to be tested, so there is less reason to expect that workers from that company would have systematically more or less positivity than other taxi drivers.

Self-selection bias could also be generated if the participants' decision to be tested is correlated with their perceived likelihood of being positive. To account for this possible bias, the survey (implemented before testing) includes questions about COVID-related symptoms and any contact with a confirmed or probable case in the past 14 days. In our main results, we exclude those with a positive answer to either of the two questions. By doing so, we account for the most likely source of bias, in a conservative way, meaning that the results should be a lower bound of the

actual positivity of the population. By excluding both symptomatic and known contacts of infected individuals, the sample is reduced to 42,164. Supplementary Table 1 shows that positivity rates are lower when excluding those individuals (3.08%) compared to when calculated on the full sample (5.75%).

To account for the over-sampling of occupations, we weight observations by the occupation population size in Bogota divided by the number of individuals of that occupation in the sample. This reweights our sample so that it is representative of Bogota as a whole. Our results, however, are not particularly sensitive to this reweighting exercise. In particular, the second row of Supplementary Table 1 shows that for the main sample used in the paper, the estimated positivity rate remains almost unchanged when reweighting (moving from 3.08 to 3.06%). The population of each occupation group comes primarily from the Gran Encuesta Integrada de Hogares (GEIH), and was completed with occupation-specific sources of information when missing. The GEIH follows the Clasificación Industrial Internacional Uniforme, which is the most broadly-used classification of occupations in Colombia. When further aggregated, occupations were pooled based on their level of economic activity, interactions, and ability to work remotely.

Since the sampling from lists is likely to be less subject to bias than the public campaign, we reproduce basic statistics and the results of Fig. 1, restricting our sample to this first group. In Supplementary Table 1, we find a (weighted) positivity rate that is somewhat higher among invited participants from the lists (3.24) compared to the one among participants from the public campaign (2.99), but the difference between the two is not significant ($p = 0.14$). As a result of the difference, the estimation of the share infected during the entire period reaches 63% when excluding participants from the public campaign as shown in Supplementary Fig. 3. In the main body of the paper we present the main results including both groups, both because this assumption is more conservative and because the estimated positivity deviates further from the NHI seroprevalence study in October when using only the list-sampled group, indicating that the inclusion of both groups is a more accurate estimate.

Running estimations without weights does not affect the main conclusions (Supplementary Fig. 2). The weights help to make the sample more representative of the Bogota population, but the results are nevertheless robust to using unweighted positivity. By contrast, keeping all individuals who have symptoms or known contact with an infected person would increase the results substantially, probably as a result of the selection bias mentioned previously (Supplementary Table 1).

Since individuals cannot be forced to take a test, self-selection, and selective acceptance will be unavoidable in observational data. But by combining multiple data sources and reducing obvious sources of bias, we are able to capture a picture of the evolution of COVID-19 a wide timeframe and a broad range of groups. Supplementary Tables 3, 4, 6, show that a wide range of socioeconomic strata, occupations, and locations were reached through this approach.

**From positivity rate to a number of infected individual per day**. The conversion from positivity to number of daily infections requires estimating the expected number of days during which a person can be tested positive when infected. In the context of this study, this depends on the sensitivity of the PCR test. Using the estimations of Miller et al.[13], we estimate this number of days to be 17. We used their results on the sensitivity of the PCR test day-by-day following the onset of the symptoms to sum the percentage of sensitivity of each day and added an extra 2 days to take into account the period prior the onset of the symptoms. We estimated that there are, on average, 17 days in which a positive person can be tested positive by a PCR test (for example 2 days at 50% sensitivity mean 1 day when the individual is detected in expected value).

To obtain the number of cases per day and per inhabitant, one needs to divide the positivity rate by 17. Intuitively, if any person that gets infected can be tested positive for 17 days on average, then the positivity rate should be 17 times higher than the numbers of daily new infections. We hence divided the positivity by 17 to obtain the number of daily cases. To illustrate this, suppose that in a population of 100 people, one new person gets infected every day, and the PCR test has a sensitivity of 100% for 17 days. Then, random testing in that population will detect a positivity rate of 17% (as long as you test at least 17 days after the first infection), because the test will be positive for all infections that occurred over the last 17 days.

We check the robustness of our estimations to variations in this 17 day figure. Estimating the expected number of days that one can be detected requires precise estimations of the sensitivity day-by-day. We found only one other study with daily test sensitivity[27], with which we estimate that, in expectation, an infected person can be tested positive during 15.5 days. Supplementary Fig. 4 replicates Fig. 1 using 15.5 instead of 17 days to calculate the total number of infected. Mechanically, a lower duration results in higher numbers of infections, because if people are detectable for a shorter period, then more infections are required to have a given level of positivity at a given point in time.

An additional work estimates the PCR sensitivity between 0 and 4 days, and between 10 and 14 days after the onset of the symptoms[28]. Their results are not comprehensive enough to enable us to estimate the total number of days in which a person can be tested positive. However, we can compare the sensitivity with the one that we use, for the same days, and find that the sensitivity is always slightly lower. Hence if the results were more complete, again we would expect to find a lower number of expected days to be tested positive, resulting in more infections. In

conclusion, the 17 days from the study that we use provide an upper bound on the expected number of days that one can be tested positive (compared with results from other studies). Thus our results provide a lower bound on actual infections, acting as a conservative estimate of the true number of infections.

**Reporting summary**. Further information on research design is available in the Nature Research Reporting Summary linked to this article.

## Data availability

The raw CoVIDA and HBS data are protected and are not available due to data privacy laws. The processed data sets are available at OPENICPSR under accession code 142121[29] (https://www.openicpsr.org/openicpsr/project/142121/).

## Code availability

The codes that support the findings of this study are available at OPENICPSR under accession code 142121[29] (https://www.openicpsr.org/openicpsr/project/142121/).

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

## Acknowledgements

The authors acknowledge generous support from the Interamerican Development Bank, the Development Bank of Latin America (CAF), the University of Los Andes and the Universidad Nacional de Colombia. Duncan Webb gratefully acknowledges support from the ED 465 at University Paris 1, and the EUR project ANR-17-EURE-0001. We thank members of the CoVIDA working group, Yessica Daniela Campaz Landazabal, Marylin Hidalgo, Juliana Quintero, Paola Betancourt, Pablo Rodríguez, Diana Sofia Rios Oliveros, Andrés Felipe Patiño, Jose David Pinzon Ortiz, Maribel Rincón, Alejandro Segura, and Leonardo Salas Zapata.

## Author contributions

R.L., G.B., and F.d.l.H. conceived the study; R.L., C.D.L.R., I.S., and D.W. analyzed the data; R.L., D.A., E.B., R.B., G.B., Z.C., A.G., L.J.H., L.L., A.R.V., S.R., N.S., and M.V. implemented the CoVIDA project, including tests and primary data collection; R.L., C.D.L.R., I.S., and D.W. wrote the paper; D.M., E.O., and R.R. provided access to HSB data and contributed to the design of the CoViDA project, all authors critically revised the various versions of the paper. This paper is part of the CoVIDA project implemented by the CoVIDA consortium, which includes the following members: E.B.[1], R.B.[1], P.B.[1], G.B.[1], Y.C.-D[1], D.C.[1], S.C.-A.[1], A.G.[1], M.G.[1], L.J.H.[1], M.H.[1], R.L.[1], L.L.[1], E.O.[1], A.R.V.[1], S.R[1], D.R[1], P.R.[1], P.R.[1], G.T.C[1], and M.V.[1].

## Competing interests

The authors declare no competing interests.
