## [Peer Review File · Nature Communications]

REVIEWER COMMENTS

Reviewer #1 (Remarks to the Author):

In the proposed paper, the authors present the results of a large study that tested many individuals for SARS-CoV-2 in Bogota. South America has been hit very hard by the COVID epidemic and obtaining an understanding of the subpopulations most at risk is much needed. In particular the paper benefits from a large sample size and it covers a wide range of different populations. Nevertheless, there are a lot of details missing from the paper and no crude results provided, which made interpreting the results difficult.

Most of the methods sections in the supplement should be moved to the main paper. At the moment it's very difficult to understand the underlying assumptions and data collection methods without digging into the supplement.

The authors use a range of different sources to obtain samples. These come from a mix of directly contacting companies as well as public campaigns. It is unclear what the mix of these samples were by type and how this changed over time. It would be useful to show the number of participants per study by month.

The paper jumps straight into modelled + smoothed results with no presentation of the raw numbers. It would be useful to present in a table with the number of tests performed, the number of positive tests by age, sex, profession etc. It would also be preferable to plot all the lines as unsmoothed to provide a more accurate presentation of the data (or at least present both raw and unsmoothed lines).

The authors assume a 17-day positivity window for the PCR test – this is a key assumption that comes from a single study. It would be useful to understand the sensitivity of this assumption to different values. Would a shorter duration (which might be expected for asymptomatic individuals) produce different results?

Figure 1 shows a smoothed curve of the cumulative proportion positive – this requires using the assumption that individuals are positive for 17 days. It would be useful to first show the raw results – so number and proportion of tests positive by month.

It would be useful to plot the positivity by region (Fig 2) also as a map – this will help readers see if it is correlated in space.

It is probably worth noting that while participants were asymptomatic at the time of testing, many may have gone on to develop symptoms. The term asymptomatic is generally taken to mean without symptoms throughout their infection.

The panels on Figure 3 would greatly benefit from having the same x-axis (and leaving the space blank where there is no data) – at the moment is difficult to compare across the panels. The authors should also avoid using the same colours across the panels in Figure 3 as there is always the temptation to compare blue with blue etc. Finally – I would avoid smoothing the curves in this Figure – it is always better to show the raw results, even if they are a bit noisier.

What source of data was used to capture the proportion of the population that had each occupation?

The proposed relationship between rise in R and the festive period seems speculative. I would downplay this. Is there potentially a role for different variants too during this time?

Similarly, the link between travellers and the wealthiest strata being hit first (line 162) also seems very speculative. I would remove.

Is the seroprevalence study data available by the different regions of Bogota? – this would provide another interesting comparison point.

Reviewer #2 (Remarks to the Author):

The authors study the RT-PCR detection of Sars-Cov2 in asymptomatic individuals in a large sample of ~ 60,000 individuals in Bogotá, Colombia. Tests were performed on large time interval (June 2020 – February 2021), corresponding to an average ~1,600 tests per week. Social economic information of the tested subjects provides a detailed landscape of the affected population by type of activity, social strata, and place of residence. Effects of imposed quarantine and the presence of two periods of larger incidence (first and second wave) are also characterized. Whenever possible, results obtained by the collected data were compared with those obtained from official reports on the Covid-19 disease provided by the health authorities.

Valuable results were obtained by the analysis of collected data. The methodology used to derive them is well described. Besides their intrinsic value, they serve for the purposes of comparison with similar results in other large LMIC cities. As a matter of fact, they may also be useful to estimate possible scenarios in cities where careful studies like this one have not been carried out.

Despite an overall positive general evaluation, the authors should work further in the manuscript to fix some inconsistencies and missing information.

1) L 50: The statement "Another half was a convenience sample based on a public invitation for free testing" characterizes how part of the tested subjects were chosen among Bogota's population. However, in the Supplementary Material, starting from line 172 in section SI.1.3.1, the term "public campaign" is used to characterize this group, without any prior definition. Please characterize this concept already in the main text.

2) Figure 3: When comparing Figures 3c and 3d, one notices that there is no shift in the position of the peak of daily deaths with respect to that of daily new cases in Figures 3c and 3d. The occurrence of such a shift has been reported for many other places. The authors should comment on this quite unexpected feature.

3) L 161: 100,000 inhabitants in Figure SI.7d -> 100,000 inhabitants in Figure 3d and Figure SI.7d

4) L 169-171: The statement "This puzzle seems to be driven by ...it remained relatively stable in the poorest strata" seems difficult to be followed. Where this information is provided? In Sup. Mat? Please inform.

Supplemental Material

5) L 127 and L 141: values of "r" should be the same

6) Figure SI.8: I notice some discrepancies between some results in the above graphs and those in Figure 2. For instance, the stay at home curve above is above that of taxi driver between June and November, but in Figure 2 this information is reversed, the value of stay at home is below that of taxi driver. Is this due to different ways of computing the rates? You use rate per 100,000 within each specific group in Figure 3 and SI.8, but percentage of the population (also within each specific group) in Figure 2. Please explain this.

7) Misspellings

MainText

a - caption Figure 2: 1st-May -> 1st-March

Supplemental Material

b - L 26: bu -> by

c - L 66: weigh -> weight

d - L 78: an -> as

e - L 99: estimation expected -> estimation of the expected

Conclusion: The presented manuscript reports relevant data on Covid-19 incidence in a very large city, providing an accurate description of the SARS-CoV-2 spread. To ensure consistency of some indicated results, text, and conclusions, a revision is necessary before a definite assessment regarding publication in Nature Communications.

Response to the reviewers of Nature Communications, for manuscript
“COVID-19 spread, detection, and dynamics in a megacity in Latin America”

Reviewer #1:

In the proposed paper, the authors present the results of a large study that tested many individuals for SARS-CoV-2 in Bogota. South America has been hit very hard by the COVID epidemic and obtaining an understanding of the subpopulations most at risk is much needed. In particular the paper benefits from a large sample size and it covers a wide range of different populations. Nevertheless, there are a lot of details missing from the paper and no crude results provided, which made interpreting the results difficult.

Most of the methods sections in the supplement should be moved to the main paper. At the moment it's very difficult to understand the underlying assumptions and data collection methods without digging into the supplement.

Thank you for this recommendation. We moved the methods section to the main manuscript, starting in page 11 of the main manuscript (section 4).

The authors use a range of different sources to obtain samples. These come from a mix of directly contacting companies as well as public campaigns. It is unclear what the mix of these samples were by type and how this changed over time. It would be useful to show the number of participants per study by month.

Thank you for this suggestion and all the other ones that contribute to a better transparency of the data that we have. We report the number of participants per study by month in Supplementary Table 2 and refer to it in line 250 of the main text.

The paper jumps straight into modelled + smoothed results with no presentation of the raw numbers. It would be useful to present in a table with the number of tests performed, the number of positive tests by age, sex, profession etc. It would also be preferable to plot all the lines as unsmoothed to provide a more accurate presentation of the data (or at least present both raw and unsmoothed lines).

We now report the number of tests performed and positivity by gender (Supplementary Table 9), by age group (Table 8) and by occupation (Table 4). Because of Nature Communications limitations on the number of tables and figures, we added these tables in the Supplementary Material, and refer to it in the main text in line 251.

Following your recommendation, in both the main document and Supplementary Material, we replaced all the graphs that had smoothed lines by unsmoothed monthly averages Figure 3 and Supplementary Figures 7 and 8).

We also added Supplementary Figure 1 which presents the raw positivity rate, month by month (and at the top of the figure the number of observations per month).

With this same intention to increase transparency of the data, in Supplementary Figure 8, we also show the number of observations per month for each occupation category. This intends to help the reader be more cautious in the interpretation in months and groups where the sample size is particularly low.

The authors assume a 17-day positivity window for the PCR test – this is a key assumption that comes from a single study. It would be useful to understand the sensitivity of this assumption to different values. Would a shorter duration (which might be expected for asymptomatic individuals) produce different results?

This is a very good point, and it is true the key results depend on this assumption. We carried out a more extensive review of the literature on this topic. We found two additional articles, of which only one (Grassly et al. 2020) has day-by-day sensitivity of the PCR test during the period following infection, which is the level of detail required to compute the expected number of days that one can be tested positive. This one leads us to an expected number of days to be detected positive that is equal to 15.5. Supplementary Figure 4, replicates the results of figure 1 using 15.5 days instead of 17. We find for example that under this assumption, by March 2021 59% of the population would have been infected. The third paper is less complete but would also result in an expected number of days that is smaller than 17. We conclude that 17 days is the highest number found across all articles, hence if this means that our main results are conservative: Our infection rates over a given period are a lower-bound compared to the infection that would be obtained with numbers from other articles; mechanically when this number of days is reduces, this implies a higher infection rate because it takes more infections to obtain the same positivity rate at a given time. This is explained in the Method Section of the main manuscript between lines 333 and 350.

Figure 1 shows a smoothed curve of the cumulative proportion positive – this requires using the assumption that individuals are positive for 17 days. It would be useful to first show the raw results – so number and proportion of tests positive by month.

We show the raw monthly positivity results in Supplementary Figure 1. Because of the space constraint, we show this in Supplementary Information.

It would be useful to plot the positivity by region (Fig 2) also as a map – this will help readers see if it is correlated in space.

Supplementary Figure 9 (which was already there) shows the cumulative share infected by region. Now we added in Supplementary Figure 10 the raw positivity, in order to be more transparent. We also present in Figure 9 a) the average Socio-economic Stratum of each Locality, to help readers see how correlated in space it is, and this correlation closely reflects correlations with our proxy for income level.

It is probably worth noting that while participants were asymptomatic at the time of testing, many may have gone on to develop symptoms. The term asymptomatic is generally taken to mean without symptoms throughout their infection.

That is very true! We made this clarification in line 37 of the main manuscript, thank you.

The panels on Figure 3 would greatly benefit from having the same x-axis (and leaving the space blank where there is no data) – at the moment is difficult to compare across the panels. The authors should also avoid using the same colours across the panels in Figure 3 as there is always the temptation to compare blue with blue etc. Finally – I would avoid smoothing the curves in this Figure – it is always better to show the raw results, even if they are a bit noisier.

We rectified figure 3 to have the same x-axis in all sub-figures. We now use a different color in panel a) compared to panels b) c) and d). These last 3 subfigures keep the same color code, in order to facilitate the reading and be consistent between colors and strata across these graphs (meaning that, for example, comparing red with red would be meaningful now, since it always represents strata 1&2).

The figure now displays non-smoothed data, averaged by month.

What source of data was used to capture the proportion of the population that had each occupation?

In lines 287-292, we clarified the source of these data. The population size of each occupation within Bogota comes mainly from the *Gran Encuesta Integrada de Hogares (GEIH)*, the primary household survey carried out by the Colombian National Administrative Department of Statistics

The proposed relationship between rise in R and the festive period seems speculative. I would downplay this. Is there potentially a role for different variants too during this time?

As recommended by the referee, we downplayed this part (lines 104-105). Given the growing importance of new variants we expanded the discussion on this topic in lines 195-208. As mentioned, the first P.1 variant in Bogota was reported on March 13th, hence it is unlikely to explain the second wave, but is likely to play a role in the third wave (after the period covered by our data).

Similarly, the link between travellers and the wealthiest strata being hit first (line 162) also seems very speculative. I would remove.

We agree and removed this sentence (line 144).

Is the seroprevalence study data available by the different regions of Bogota? – this would provide another interesting comparison point.

We asked our colleagues from the National Health Institute, but unfortunately this data is not available at the moment.

We are very grateful for the thoughtful comments, extremely helpful and allowing us to have a version that increases transparency and understanding of the reader.

Reviewer #2 (Remarks to the Author):

The authors study the RT-PCR detection of Sars-Cov2 in asymptomatic individuals in a large sample of ~ 60,000 individuals in Bogotá, Colombia. Tests were performed on large time interval (June 2020 – February 2021), corresponding to an average ~1,600 tests per week. Social economic information of the tested subjects provides a detailed landscape of the affected population by type of activity, social strata, and place of residence. Effects of imposed quarantine and the presence of two periods of larger incidence (first and second wave) are also characterized. Whenever possible, results obtained by the collected data were compared with those obtained from official reports on the Covid-19 disease provided by the health authorities.

Valuable results were obtained by the analysis of collected data. The methodology used to derive them is well described. Besides their intrinsic value, they serve for the purposes of comparison with similar results in other large LMIC cities. As a matter of fact, they may also be useful to estimate possible scenarios in cities where careful studies like this one have not been carried out.

Despite an overall positive general evaluation, the authors should work further in the manuscript to fix some inconsistencies and missing information.

1) L 50: The statement “Another half was a convenience sample based on a public invitation for free testing” characterizes how part of the tested subjects were chosen among Bogota’s population. However, in the Supplementary Material, starting from line 172 in section SI.1.3.1, the term “public campaign” is used to characterize this group, without any prior definition. Please characterize this concept already in the main text.

Now early-on, in lines 43-44 we introduce the term and explain it: “The second half was a convenience sample based on our “public campaign”. Upon request of the other referee, the Methods Section was now added to the main manuscript. Hence now in lines 247-249 and lines 291-301 one can find additional explanations and robustness check with respect to this sample. 44 widely disseminated public invitation for free testing.

2) Figure 3: When comparing Figures 3c and 3d, one notices that there is no shift in the position of the peak of daily deaths with respect to that of daily new cases in Figures 3c and 3d. The occurrence of such a shift has been reported for many other places. The authors should comment on this quite unexpected feature.

Thank you for your careful look at the data and figures. The reason why we do not have this shift in our data is that even for daily deaths, we use the date of the test rather than the date of the death. We clarified this in the legend of figure 3.

3) L 161: 100,000 inhabitants in Figure SI.7d -> 100,000 inhabitants in Figure 3d and Figure SI.7d

Thank you, we corrected the typo.

4) L 169-171: The statement “This puzzle seems to be driven by ...it remained relatively stable in the poorest strata” seems difficult to be followed. Where this information is provided? In Sup. Mat? Please inform.

Indeed, we forgot to mention that this information is provided in table 3. We now mention it in line 158, thanks!.

Supplemental Material

5) L 127 and L 141: values of “r” should be the same

Thank you for noticing! Indeed you are right and we made the correction. Now in lines 16 and 29 of Supplementary Material.

6) Figure SI.8: I notice some discrepancies between some results in the above graphs and those in Figure 2. For instance, the stay at home curve above is above that of taxi driver between June and November, but in Figure 2 this information is reversed, the value of stay at home is below that of taxi driver. Is this due to different ways of computing the rates? You use rate per 100,000 within each specific group in Figure 3 and SI.8, but percentage of the population (also within each specific group) in Figure 2. Please explain this.

Again, thanks for this very neat detection of details! Following the comments of the first referee, we replaced all figures with smoothed representations of the data by non-smoothed monthly averages, and in some figures, including Supplementary Figure 8, we document the number of observations per month, separately for each occupation. As you can notice in Supplementary Figure 8, the high positivity of “stay at home mothers” in July is driven by only 7 observations, compared to much more observations in the following months, hence this ends up having a very small weight when the positivity is calculated over a longer time period.

Thank you, your comment helped us think of a way to present the figures that avoids paying too much attention to months with small number of observations and raises puzzles such as the one you mentioned. We hope that this new presentation helps solve not only this inconsistency but any other one that could appear because of our previous formatting with smoothed values and lack of information about the number of cases per month.

7) Misspellings

MainText

a - caption Figure 2: 1st-May -> 1st-March

Supplemental Material

b - L 26: bu -> by

c - L 66: weigh -> weight

d - L 78: an -> as

e - L 99: estimation expected -> estimation of the expected

Thank you very much, we corrected all the typos mentioned (or in some cases the sentence just disappeared with the recent changes).

Conclusion: The presented manuscript reports relevant data on Covid-19 incidence in a very large city, providing an accurate description of the SARS-CoV-2 spread. To ensure consistency of some indicated results, text, and conclusions, a revision is necessary before a definite assessment regarding publication in Nature Communications.

Many thanks for your careful review, which helped us detect mistakes and improve the clarity of the document!

REVIEWERS' COMMENTS

Reviewer #1 (Remarks to the Author):

The authors have been responsive to my concerns. I have no further comments. I congratulate them on the work - it must have been an enormous undertaking